# A Crucial Role for Ergosterol in Plasma Membrane Composition, Localisation, and Activity of Cdr1p and H^+^-ATPase in *Candida albicans*

**DOI:** 10.3390/microorganisms7100378

**Published:** 2019-09-22

**Authors:** Jakub Suchodolski, Jakub Muraszko, Przemysław Bernat, Anna Krasowska

**Affiliations:** 1Department of Biotransformation, Faculty of Biotechnology, University of Wroclaw, 50-383 Wrocław, Poland; jakub.suchodolski@uwr.edu.pl (J.S.); jakub.muraszko@gmail.com (J.M.); 2Department of Industrial Microbiology and Biotechnology, Faculty of Biology and Environmental Protection, University of Łódź, 90-237 Łódź, Banacha 12/16, Poland; przemyslaw.bernat@biol.uni.lodz.pl

**Keywords:** *Candida albicans*, lanosterol 14α-demethylase, *ERG11*, ergosterol, Cdr1p, H^+^-ATPase

## Abstract

*Candida albicans* is an opportunistic fungal pathogen of humans. Treatment of *C. albicans* infections relies on azoles, which target the lanosterol 14α-demethylase (Erg11p) encoded by the *ERG11* gene. Our results show that targeted gene disruption of *ERG11* can result in resistance to ergosterol-dependent drugs (azoles and amphotericin B), auxotrophy and aerobically viable *erg11Δ/Δ* cells. Abnormal sterol deposition and lack of ergosterol in the *erg11Δ/Δ* strain leads to reduced plasma membrane (PM) fluidity, as well as dysfunction of the vacuolar and mitochondrial membranes, resulting respectively in defects in vacuole fusion and a reduced intracellular ATP level. The altered PM structure of the *erg11Δ/Δ* strain contributes to delocalisation of H^+^-ATPase and the Cdr1 efflux pump from the PM to vacuoles and, resulting in a decrease in PM potential (Δψ) and increased sensitivity to ergosterol-independent xenobiotics. This new insight into intracellular processes under Erg11p inhibition may lead to a better understanding of the indirect effects of azoles on *C. albicans* cells and the development of new treatment strategies for resistant infections.

## 1. Introduction

The fungal microorganism *Candida albicans* is a part of the healthy mucosal surface microbiota of around 50% of the human population [1]. It is also the cause of fungal infections in millions of immunologically deficient individuals worldwide, causing candidiasis of oral, gastrointestinal, and vaginal surfaces, as well as candidemia or invasive candidiasis [2]. Current treatment of candidiasis involves the use of antifungal agents, such as polyenes and azoles. Both types of antifungals cause disturbances in the structure and functioning of the plasma membrane, either by binding ergosterol (polyenes, e.g., amphotericin B) or by inhibiting the enzyme lanosterol 14α-demethylase, which is involved in ergosterol biosynthesis (azoles, e.g., fluconazole) [3].

Ergosterol is a neutral lipid of fungal membranes, and is critical for many cellular processes [4], hence disruption of its synthesis has become a focus of antifungal therapies, particularly as the prevalence of *Candida* resistance to many antifungal drugs increases. The lanosterol 14α-demethylase gene (*ERG11*) encodes a member of the cytochrome P450 family of enzymes that converts lanosterol to ergosterol and is therefore essential for the synthesis of ergosterol [5]. Alterations to the target enzyme, either via overexpression or point mutations of the *ERG11* gene, are a cause of growing azole resistance in *C. albicans* [6]. Another mechanism for the development of azole resistance involves overexpression of genes encoding multidrug resistance (MDR) transporters, which pump out drugs from the cell. There are three MDR transporters involved in azole efflux in *C. albicans*: Cdr1p and Cdr2p, which belong to the ATP-Binding Cassette (ABC) family, and Mdr1p, which belongs to the Major Facilitator Superfamily (MFS) [7].

Recently, the first results have been published which point to other important functions of *ERG11* in *C. albicans*, amongst others, including oxidative stress adaptation, hyphal elongation, and virulence in vivo [8]. Our study shows that the expression of *ERG11* is necessary to Cdr1p activity. In an *ERG11* knockout *C. albicans* strain, we also observed reduced H^+^-ATPase activity and its delocalization to the vacuole, reduced membrane potential and reduced intracellular ATP. To our knowledge, no such results have been published so far. Based on these results, blocking of expression of *ERG11* gene appears to be a potential way for antifungal therapy. Thus, a factor that would reduce or block *ERG11* gene expression could reduce the pathogenicity of *C. albicans*.

## 2. Materials and Methods

### 2.1. Chemicals

Chemicals and reagents used in this study were purchased from the following sources: sodium dodecyl sulfate (SDS), 2-deoxy-d-glucose, fluconazole; ketoconazole; conventional amphotericin B, rhodamine 6G (R6G), laurdan, fluphenazine, β-mercaptoethanol (BME), albumin fraction V (BSA), ethylenediaminetetraacetic acid (EDTA), trichloroacetic acid (TCA), ergosterol, lanosterol, cholesterol, and BSTFA/TMCS (N,O-bis(trimethylsilyl) trifluoroacetamide/trimethylchlorosilane) (Sigma-Aldrich; Poznań, Poland); commercial antibodies (HRP-conjugated goat anti rabbit (manufacturer: GE Healthcare; distributor: Sigma-Aldrich; Poznań, Poland), mouse monoclonal anti-GFP (manufacturer: Roche; distributor: Sigma-Aldrich; Poznań, Poland), HRP-conjugated rabbit anti mouse (manufacturer: GE Healthcare; distributor: Sigma-Aldrich; Poznań, Poland); d-glucose, bacteriological agar, propidium iodide (PI), zymolyase, D-sorbitol, brefeldin A (manufacturer: Bioshop; distributor: Lab Empire; Rzeszów, Poland); peptone, yeast extract (YE) (manufacturer: BD; distributor: Diag-med; Warszawa, Poland); chloroform (CHCl3), methanol (MetOH), acetone (Chempur; Piekary Śląskie, Poland); KOH and HCl (Avantor; Gliwice, Poland); hexane (manufacturer: JT Baker; distributor: Avantor; Gliwice, Poland); pyridinium, 4-(2-(6-(dibutylamino)-2-naphthalenyl)ethenyl)-1-(3-sulfopropyl), hydroxide inner salt (di-4-ANEPPS), restriction enzymes (*Apa*I, *Not*I, *Sac*I and *Xho*I), FM 4-64 dye (Thermo Fisher; Warszawa, Poland); generic rabbit polyclonal anti-Cdr1p antibodies were a gift from Prof. D. Sanglard (Lausanne, Switzerland). All chemicals were high purity grade.

### 2.2. Strains and Growth Conditions

The *C. albicans* strains used in this study are listed in Table 1. CAF2-1 was a kind gift from Prof. D. Sanglard (Lausanne, Switzerland). YHXW11 was a kind gift from Prof. J. Konopka (Stony Brook, USA). ASCa1 was previously constructed by our team. KS014, KS018, KS020, KS021, KS023, KS028, KS043, KS044 and KS045 were constructed during this study. Strains were pre-grown at 28 °C on YPD medium (2% glucose, 1% peptone, 1% YE) in a shaking incubator (120 rpm). Agar in a final concentration of 2% was used for medium solidification.

Growth phases were determined in 20 mL YPD medium (28 °C; shaking: 120 rpm; starting OD_600_ = 0.1) with OD_600_ measurements performed using a Hach Odyssey DR/2500 spectrophotometer. Doubling times (*t_d_*s) were calculated according to the following formula:
$$tdS=\;\frac{Duration\;of\;growth\;\left[h\right]\times log\left(2\right)}{\log\left(A600\;t=0\right)-log\left(A600\;t=h\right)}.$$

For specific experiments, cells were grown until they reached either early (8 h), late (14 h) logarithmic or stationary (24 h) phase. Cells were centrifuged (4500 rcf, 5 min.), washed twice (4500 rcf, 5 min.) with either phosphate-buffered saline (PBS), H_2_O_dd_ or citrate-phosphate (CP) buffer (pH 6.0), and resuspended in either PBS, H_2_O_dd_ or CP to the indicated OD_600_.

### 2.3. Plasmids and Strains Construction

Plasmid pSFS5 was a generous gift from Prof. J. Morschhäuser (Würzburg, Germany). The *Apa*I–*Xho*I upstream fragment of the *ERG11* gene (positions −607 to −1 bp) was amplified from CAF2-1 genomic DNA with the primer pair E11U_F (5′-TGCGGGCCCGATTACAATCAACTTGAATATTAC-3′; *Apa*I underlined) and E11U_R (5′-CGGCCTCGAGATTGAGTTATGATCTTCTTG-3′; *Xho*I underlined). The fragment was cloned into the *Apa*I-*Xho*I site of the pSFS5 plasmid to generate plasmid pSFS5E11U. The *Not*I-*Sac*I *ERG11* downstream fragment (positions +1593 to +2074 bp) was amplified with primer pair E11D_F (5′-TAAACTATGCGGCCGCCGGCAACTTTCTTTCGATTC-3′; *Not*I underlined) and E11D_R (5′-GCGCGAGCTCGCATGGTGAGGATCTGATG-3′; *Sac*I underlined). This fragment was then cloned into the *Not*I-*Sac*I site of the pSFS5E11U to generate pSFS5E11.

*C. albicans* strains were transformed by electroporation [12] with the linear gel-purified *Apa*I-*Sac*I fragment from pSFS5E11. The presence of the *SAT1* marker was verified using primer pair SAT1_F (5′-CGGATGATGACTCTGATGAAG-3′) and SAT1_R (5′-GGACCACCTTTGATTGTAAATAG-3′). The correct integration of the construct into genomic loci was verified using primers SAT1_F and E11D_R2 (5′-GCATGGTGAGGATCTGATG-3′). Site-specific (Flp-FRT) recombination in heterozygous (*ERG11/erg11Δ::SAT1-FLIP)* strains was induced by BSA and verified using primers E11U_F2 (5′-GAATATTACTGGATGCAGAACAACA-3′) and E11D_R2. *ERG11/erg11Δ::FRT* strains were transformed as above, and homozygous genotypes (*erg11Δ::SAT1-FLIP/erg11Δ::FRT)* were verified using E11_F (5′-TAGGGGTTCCATTTGTTTACAAC-3′) and E11_R (5′-CATTGGCAACCCCATGAG-3′); E11M_F (5′-TAGATGGGATACTGCTGCTG-3′), and the E11D_R2; E11U_F2 and E11_R primer pairs.

### 2.4. Phenotypic Tests and MIC_50_ Determination

KS028 and CAF2-1 strain suspensions (PBS; OD_600_ = 0.7; prepared from overnight cultures) were serially diluted up until 10^−3^. 2 µL of each dilution, from 10^0^ to 10^−3^, were spotted onto YPD or YNB-based agar and cultured 48 h at 28 °C. Plates were photographed using a FastGene^®^ B/G GelPic imaging box (Nippon Genetics; distributor: Abo, Gdańsk, Poland) 

MIC_50_ values of azoles and amphotericin B towards KS028 and CAF2-1 were determined, according to CLSI, with modifications described in [13].

### 2.5. Real Time PCR

The RNA isolation, cDNA synthesis and calculations of gene expression levels were prepared as previously described [10]. The following gene-specific primers were used: RDN18F (5′-AGAAACGGCTACCACATCCAA-3′), RDN18R (5′-GGGCCCTGTATCGTTATTTATTGT-3′), ERG11F (5′-TTTGGTGGTGGTAGACATA-3′), ERG11R (5′-GAACTATAATCAGGGTCAGG-3′), ERG3F (5′-CCATCATGAATCATGACAGTCC-3′), ERG3R (5′-TGCTTCTCATGCTTTCCATC-3′), PMAF1 (5’-TTCTGAAATTGCTGATTTCGTTG-3′), PMAR1 (5’-CGTTAACACCATCACCAGTC-3′), CDR1F (5′-TTTAGCCAGAACTTTCACTCATGATT-3′), CDR1R (5′-TATTTATTTCTTCATGTTCATATGGATTGA-3′).

### 2.6. Isolation of Plasma Membranes

The method was based on [14], with modifications. KS028 and CAF2-1 suspensions (PBS; concentrated to OD_600_ = 20) were pelleted and resuspended in lysis medium (1 M sorbitol, 0.1 M EDTA, 1% BME, 3 mg/mL zymolyase). After incubation (30 min.; 37 °C), protoplasts were washed three times with 1.2 M sorbitol, lysed by ice-cold H_2_O_dd_ shock and additionally broken by sonication (2 min. in 5 s cycles; ice bath) using an ultrasonic processor (Vibra-cell VCX 130, Sonics; distributor: VWR, Gdańsk, Poland). Cell lysate was centrifuged (10,000 rcf; 10 min; 4 °C) and the supernatant ultracentrifuged (100,000 rcf; 60 min., 4 °C) using a Micro Ultracentrifuge CS150FNX (Hitachi; distributor: Accela, Prague, Czech Republic). The crude plasma membrane pellets were resuspended in saline solution and CHCl_3_-MetOH (1:2, vol/vol) was added. After vigorous stirring for 16 h at 4 °C, the CHCl_3_ layer was concentrated using nitrogen gas.

### 2.7. Sterol Analysis

Sterol analysis was performed according to the method proposed by [15]. To the evaporated lipid samples, 0.5 mL CHCl_3_, 0.5 mL MetOH-KOH (0.6 M) and 20 µg of cholesterol were added. After 1 h of incubation (23 °C) 0.325 mL 1M HCl and 0.125 mL H_2_O_dd_ were added and centrifuged (5000 rcf; 10 °C; 5 min). The lower layer was transferred to fresh tubes, dried and 100 μL BSTFA/TMCS was added. Samples were heated at 85 °C for 90 min, then 50 µL hexane was introduced to tubes and vortexed. The analysis was performed with a gas chromatograph (Agilent 7890) equipped with column HP 5 MS (30 m × 0.25 mm inner diameter, i.d. × 0.25 mm film thickness, f.t.) and a 5975C Mass Detector. The column was maintained at 100 °C for 0.5 min^−1^, then increased to 240°C at a rate of 25 °C min^−1^, and finally to 300 °C at a rate of 3 °C min^−1^ (for 5 min) with helium as a carrier gas at a flow rate of 1 mL·min^−1^ [15]. The injection port temperature was 250 °C. Cholesterol was used as an internal standard. Tetramethylsilane (TMS)-derived ergosterol and lanosterol were analysed with reference to retention times and fragmentation spectra for standards. Other sterol TMS ethers were identified by comparison with the NIST database or literature data and quantitated using a standard curve for lanosterol.

### 2.8. Membrane Fluidity Assessment

The assay was based on a *Saccharomyces cerevisiae* protocol [16], with modifications. KS028 and CAF2-1 suspensions (PBS, OD_600_ = 0.1, 3 mL) were labelled with laurdan (final conc. = 5 × 10^−6^ M; 20 min.; 25 °C; in darkness). The probes were excited at 366 nm (Ex slit = 10 nm), and fluorescence spectra were recorded at 400–550 nm (Em slit = 2.5 nm) (PMT voltage = 400 V) using a fluorescence spectrophotometer equipped with a xenon lamp (HITACHI F-4500; manufacturer: Hitachi, Tokyo, Japan). For the data analysis, modified general polarisation (GP) was calculated as follows: the difference of the sum of fluorescence intensities (IFs) from 425 to 450 nm and the sum from 475 to 525 nm, divided by the sum of IFs from 425 to 450 nm and from 475 to 525 nm.

### 2.9. Microscopic Studies

Strains ASCa1, KS023, YHXW11 and KS045 were suspended in PBS, concentrated and observed under a Leica SP8 LSM microscope (Leica microsystems; Wetzlar, Germany) for Cdr1-GFP and Pma1-GFP localization study. Vacuolar staining with FM 4-64 dye was prepared according to [17].

### 2.10. Proton Extrusion Assay

The method was based on a *S. cerevisiae* protocol [18], with modifications. Real time acidification of KS028 and CAF2-1 strain suspensions (H_2_O_dd_; OD_600_ = 1.0; 20 mL) was monitored every 10 s for 12 min using a pH-meter (Eutech Instruments CyberScan PH 5500, ThermoFisher Scientific, Warsaw, Poland), equipped with MiniTrode electrode (manufacturer: Hamilton; distributor: Sigma-Aldrich; Poznań, Poland). Due to exposure to hypotonic conditions, cell suspensions under identical conditions were checked for plasma membrane permeabilisation (using PI, according to the protocol of [13]). In each experiment, pH values at t_0_ were equal to 5.7 ± 0.1. For clearer presentation these have been normalised to 5.7.

### 2.11. Di-4-ANEPPS Assay

KS028 and CAF2-1 suspensions (CP; OD_600_ = 0.1; 3 mL) were labelled with di-4-ANEPPS according to our protocol [19]. For data analysis, red-blue signal ratio (R-B ratio) was calculated by dividing the sum of IFs between 580 and 620 nm by the sum of IFs between 540 and 580 nm as described previously [19]. All results were normalised to 1 for the plasma membrane potential of 8 h CAF2-1 cells

### 2.12. ATP Measurements

Intracellular ATP was measured according to our protocol [20], with modifications. KS028 and CAF2-1 suspensions (H_2_O_dd_; OD_600_ = 0.2) were mixed with lysis buffer and luciferase substrate (BacTiter-Glo microbial cell viability assay, Promega, Mannheim, Germany), according to the manufacturer’s instructions. After 15 min incubation in room temperature (RT) the suspensions were read with a microplate luminometer (EG&G BERTHOLD MicroLumat LB 96P, Berthold Technologies, Bad Wildbad, Germany). RLU/s (relative light units per second) were normalised to 1 for 8 h CAF2-1 cells.

### 2.13. Western Blotting

Crude protein extracts from CAF2-1, KS028, YHXW11 and KS045 strains were isolated according to [21], with the following modifications: after TCA precipitation, proteins were washed with ice-cold acetone, dried and resuspended in SB followed by 45 min denaturation at 37 °C. Electrophoretic separation, transfer and immunodetection of Cdr1p (CAF2-1, KS028) was performed following [21]. For Pma1p detection, the following modifications were applied: crude proteins from YHXW11 and KS045 strains were separated on 10% SDS-polyacrylamide gels and primary mouse anti-GFP antibodies were used, followed by HRP-conjugated rabbit antimouse antibodies. The remaining steps were performed as in [21].

### 2.14. R6G Efflux Assay

The R6G efflux assay was performed as described previously [21]. IFs were collected 15 min after R6G efflux initiation and normalised to 1 for the efflux activity of 8 h CAF2-1 cells.

### 2.15. Statistical Analysis

For each experiment we performed at least three independent replicates. Statistical significance was determined using Student’s *t*-test (binomial, unpaired).

## 3. Results

### 3.1. C. albicans Strain erg11Δ/Δ Is Auxotrophic, Resistant to Azoles and Amphotericin B, and Viable under Aerobic Conditions

To investigate the influence of Erg11p inhibition on drug resistance mechanisms, *SAT1* flipper targeted gene disruption [22] was used to generate the following *erg11Δ/Δ* homozygous knock-outs: WT *erg11Δ/Δ* (KS028); *PMA1-GFP erg11Δ/Δ* (KS045); and *CDR1-GFP erg11Δ/Δ* (KS023). All three strains displayed identical growth phenotypes. Most of the subsequent presented experiments (excluding microscopic observations) were performed in the KS028 strain.

Cultured in complex YPD medium, the KS028 (*erg11Δ/Δ*) strain was viable under aerobic conditions with no differences in the temporal distribution of growth phases from the WT strain (Figure 1A). However, the growth rate of the *C. albicans* K028 strain was significantly (*p* = 8.7 × 10^−5^) lower than that of the WT strain (t_d_s = 179.4 ± 4 min and 207.4 ± 9.4 min for WT and KS028 strains, respectively).

Cultured on solid complex medium (YPD), the KS028 strain developed visually sparser groupings of colonies than the WT strain (Figure 1B). The *C. albicans* KS028 strain displayed an auxotrophic phenotype due to the lack of growth on solid minimal medium (YNB) (Figure 1B). Therefore, the complex solid medium was chosen for the phenotypic tests with antimycotics. We used complex broth for all other experiments.

The *C. albicans* strain KS028 showed no growth inhibition within the tested range of amphotericin B (0.125 to 10 µg/mL), fluconazole (0.5 to 128 µg/mL), and ketoconazole (0.05 to 8 µg/mL) (Figure 1B). Similarly, the resistance of KS028 strain towards azoles and amphotericin B was expressed as no detectable MIC_50_ values of these drugs. The growth of the WT strain was, however, fully inhibited by amphotericin B from levels of 0.5 µg/mL (MIC_50_ = 0.25 µg/mL). The residual growth of the WT strain was present but limited at concentrations from 1 µg/mL fluconazole and 0.05 µg/mL ketoconazole upwards (MIC_50_ = 1 and 0.015 µg/mL for fluconazole and ketoconazole, respectively). According to literature, this can be attributed to the fungistatic properties of azoles [23]. The growth of the WT strain treated with azoles (≥1 µg/mL fluconazole and ≥0.05 µg/mL ketoconazole, Figure 1B and data not shown) was slower than that of KS028. On YPD with and without a 0.5 µg/mL fluconazole addition, the growth of both strains was comparable (Figure 1B).

### 3.2. The C. albicans Strain erg11Δ/Δ Accumulates Lanosterol at a Constant Level during Growth, but 14α Methylergosta-8-24(28)dienol at Increasing Level due to High Expression of ERG3 Gene

The inhibitory effect of azoles on Erg11p is dependent on *C. albicans* growth phase [24,25]. Thus we examined the physiology of *erg11Δ/Δ* knock-outs throughout the process of cell aging. The following time points were chosen: early and late logarithmic phase (8 and 14 h, respectively), and stationary phase (24 h). 

*ERG11* and *ERG3* gene expression in *C. albicans* WT after 14 and 24 h culture was considerably lower than after 8 h culture (Figure 2). After 8 h culture, the *ERG11* knockout strain showed four-fold higher expression of *ERG3* than the WT strain. After 14 h culture, in *C. albicans* KS028, the expression of *ERG3* was less than half that of early logarithmic phase (8 h), and remained at a comparable level after 24 h (Figure 2B). In the WT strain only, we compared 2^−∆∆CT^ values for the expression of the *ERG3* and *ERG11* genes. After 8, 14, and 24 h growth, expression of the *ERG3* gene was 2, 4, and 1.5-fold lower than the expression of the *ERG11* gene, respectively.

Blocking the synthesis of ergosterol by inhibition of the Erg11p results in the accumulation of lanosterol, which is converted by ∆(24)-sterol C-methyltransferase (Erg6p) into eburicol, then by Δ5,6-desaturase (Erg3p) into 14α methylergosta-8-24(28)dienol [26]. In the strain with uninterrupted ergosterol synthesis, we observed the highest ergosterol accumulation during logarithmic growth (14 h). After 24 h, we observed almost a 10-fold decrease in ergosterol from 14 h levels (Table 2). As expected, we did not detect 14α methylergosta-8-24(28)dienol in *C. albicans* WT. The level of lanosterol decreased with the aging of the culture, and after 24 h it was less than half the level of ergosterol (Table 2). In the *C. albicans* KS028 strain lacking the *ERG11* gene, the level of lanosterol was stable, regardless of the age of culture, whilst the level of 14α methylergosta-8-24(28)dienol increased with the aging of the culture and after 24 h culture was the highest (Table 2). In addition, small levels of eburicol, 14α methylfecosterol and 4α methylfecosterol were determined in the strain KS028 (Appendix A).

The fluidity of the plasma membrane (PM) depends in part on the concentration and type of sterols expressed [27]. Comparing *C. albicans* WT with KS028 strains, we observed a significantly lower PM fluidity in the strain without ergosterol (Table 2), which indicates the significant role of ergosterol in the construction and functionality of the membrane. In both strains, membrane fluidity decreased with the aging of the culture, and in stationary phase (24 h), was accordingly 4.5- and 2.5-fold lower for the WT and KS028 strains than in late logarithmic phase (14 h) (Table 2).

### 3.3. In C. albicans Strain erg11Δ/Δ, ATP Concentration, H^+^-ATPase Activity and Plasma Membrane Potential Are Reduced, and Pma1p Delocalised Earlier to Vacuoles

Ergosterol is the most abundant sterol in cell membranes, being present not only in the plasma membrane (PM), but also in the membranes of other organelles, such as vacuoles or mitochondria [4,28]. In *S. cerevisiae*, ergosterol deficient mutants have defects, such as in vacuole fusion and mitochondrial morphogenesis [4]. We noticed many small vacuoles in *C. albicans* KS028 cells by the 14 h of culture in contrast to the WT strain in which this effect was observed only in stationary phase (Figure 3 and Figure 4). Ergosterol deficiency, caused by the action of azoles on *C. albicans*, contributes to the abnormal functioning of these organelles and a reduced level of intracellular ATP [29]. Comparing the amount of ATP in *C*. *albicans* WT and KS028, we found a slightly lower level of ATP in the non-ergosterol strain at the late logarithmic phase (14 h) and almost five-fold lower levels of ATP in this strain in stationary phase (24 h) (Table 3).

The lack of ergosterol in the PM as well as a reduced level of ATP in the cell may cause, as demonstrated previously in *Candida glabrata*, the abnormal functioning of H^+^-ATPase [30], an enzyme responsible for maintaining membrane potential [31]. In *C. albicans* KS045 strain, which lacks ergosterol, and with GFP-labelled H^+^-ATPase, we observed the delocalisation of Pma1p (H^+^-ATPase) from the PM to the vacuole after 8 h of culture in contrast to the WT strain, in which Pma1p after 8 h of strain growth was still localised in the PM (Figure 3A). In both tested strains, Pma1p was also present in the correct place of its functioning, i.e., in the PM after up to 24 h of culturing (Figure 3A). 

The activity of H^+^-ATPase in real time can be measured by acidification of the cell environment [18]. We observed, together with the aging of the culture, a decrease in the pH of the surroundings of the cells. The medium containing the *C. albicans* WT strain after 12 min of measurement was about 0.3 pH units lower than the medium of the KS028 strain in the early logarithmic phase of growth (8 h) and respectively about 0.2 pH units lower in the late logarithmic phase of growth (14 h) (Figure 3B). Differences between strains in acidification of their external environments indicate a significantly reduced H^+^-ATPase activity in the strain without ergosterol compared to the WT strain. After 14 h of culture, in the KS028 strain, H^+^-ATPase showed very weak activity (Figure 3B). This effect might have also contributed to the lower level of Pma1p in the KS028 strain than in the WT strain, regardless of the phase of growth (Figure 3D). This low level of Pma1p results from decreased *PMA1* gene expression (Figure 3E). In both strains, *PMA1* gene expression decreased with cell aging, however after 14rs and 24 h of growth, gene expression was sufficient to maintain a stable level of Pma1p.

The plasma membrane potential in the KS028 strain was lower compared to the WT strain, but it was not as reduced, as indicated by the weak H^+^-ATPase activity measured by acidification of the environment outside the cells (Figure 3C).

### 3.4. In C. albicans Strain erg11Δ/Δ, Cdr1p Is Still Synthesised during Growth but Has Reduced Activity and Is Rapidly Delocalised to the Vacuole

Ergosterol, as a sterol present in fungi and absent in mammalian cells, is a target for antifungal drugs such as azoles [32]. Azoles, especially fluconazole, are removed from *C. albicans* cells by PM-localised ABC and MFS transporters [33]. Any lack or deficiency of ergosterol in PM may result in dysfunction of membrane proteins, including ABC transporters through their delocalisation to the inside of the cell. As in the case of Pma1 (Figure 3A), in a ergosterol knockout strain we observed the delocalisation of Cdr1p from the PM to the vacuole by the early logarithmic phase of growth (8 h) in contrast to the WT strain, which maintained the correct localisation of this protein in the PM after 8 h culture (Figure 4A). Results obtained in Western blot analysis and Real-time PCR showed that Cdr1p was still synthesised during the growth of KS028. Additionally, the *CDR1* gene underwent increased expression in KS028, in contrast to the WT strain in which we did not observe such processes (Figure 4B,C). Despite the additional level of Cdr1p in the KS028 strain, the activity of this transporter measured by the standard R6G efflux test was more than half of that of the WT strain (Figure 4D). In contrast to the WT strain, *C. albicans* KS028 showed high sensitivity to Brefeldin A and Fluphenazine, which are substrates for Cdr1p and have toxicity mechanisms not related to membrane interaction [34] (Figure 4E). This result indicates a disturbed Cdr1p activity and confirms the reduced activity of the Cdr1 transporter in the strain lacking ergosterol, as measured with the R6G test (Figure 4D,E).

## 4. Discussion and Conclusions

The *ERG11* gene is considered essential to *C. albicans* viability [35], despite Erg11p blockage by azoles having a fungistatic rather than fungicidal effect [23]. Sanglard et al. [36] showed that *C. albicans erg11Δ/Δ* mutants could be produced from either an *erg3Δ/Δ* background or as a result of mitotic recombination after culturing *ERG11/erg11Δ* heterozygotes in the presence of amphotericin B.

In this work, we show that the targeted gene deletion of *ERG11* can be achieved using selection on complex YPD medium (here YPD with nourseothricin). We identified the auxotrophic phenotype of the *erg11Δ/Δ* strain (Figure 1B, YNB), which is suppressed after adding the combination of adenine and uracil, but not amino acids (Appendix A). Thus the potential for selecting *erg11* transformants on minimal media could have previously been restricted.

In contrast to reported *erg11*Δ mutants of *S. cerevisiae* and *C. glabrata* [35,37,38], *C. albicans erg11Δ/Δ* was viable under aerobic conditions (Figure 1A), most likely due to intraspecies differences in sterol metabolism under aerobic and anaerobic conditions [39]. *S. cerevisiae* and *C. glabrata* import exogenous sterols under hypoxia and are characterised by aerobic sterol exclusion, whereas, *C. albicans* is able to assimilate sterols only under aerobic conditions, and therefore relies mostly on endogenous ergosterol [39].

Due to the lack of target enzyme (Erg11p) and the final product of the pathway (ergosterol), the *C. albicans erg11Δ/Δ* strain was invulnerable towards azoles and amphotericin B (Figure 1B), which is in agreement with data previously published by Sanglard et al. [36]. We observed differences between *C. albicans erg11Δ/Δ* and WT strains in type and level of sterol expression (Table 2). We observed a four-fold increase in the expression of *ERG3* in *erg11Δ/Δ* compared with the WT strain, which is consistent with the conversion of 14α methylfecosterol to 14α methylergosta-8-24(28)dienol [40]. 

In spite of the fact that 14α methylergosta-8-24(28)dienol is considered toxic to cells, in *C. albicans* KS028, constant expression of *ERG3* is maintained, even in stationary phase (Figure 2). We noted a greater restriction of growth in the *C. albicans erg11Δ/Δ* with *erg3Δ/Δ* background (DSY1764 strain) [36] than in KS028 (*erg11Δ/Δ* single knock-out) (Appendix A). Thus, the toxic effect of 14α methylergosta-8-24(28)dienol as a result of *ERG3* expression during Erg11p inhibition may be unique to *S. cerevisiae,* in which the phenotype was originally described [41]. 

Sterols, especially ergosterol, are necessary for proper PM construction and fluidity, however, our results indicate that PM fluidity also depends on other factors, especially in cells in stationary phase (Table 2). Qi Y. and co-workers found that, in *C. glabrata*, the fluidity of the PM as well as the activity of H^+^-ATPase decreases with decreasing extracellular pH [30]. When cells reach stationary phase, the environment is rich in metabolites, including organic acids, and the pH is lower than at the start of the culture, which may explain the strong decrease in PM fluidity we observed following 24-h culture of both strains (Table 2). Neither the replacement of ergosterol with 14α methylergosta-8-24(28)dienol, nor increased lanosterol in the KS028 strain, restored for the fluidity of the PM to that of the WT strain, therefore the PM in the strain without ergosterol was more rigid (Table 2). Sterols accumulate mostly in the inner leaflet of the PM [42], whereas FM4-64 dye is inserted into the outer leaflet of the PM and, from there, is passed on to vacuolar membranes by endocytosis [43]. We observed partial accumulation of the dye in the PM in the *erg11Δ/Δ* strain, contrary to the WT strain, where the dye was entirely accumulated in the vacuolar membranes (Figure 3A and Figure 4A). Increased rigidity of the PM in *erg11Δ/Δ* strain most likely increases the duration of outer-to-inner leaflet transport, as shown for the FM4-64 dye.

Despite a lack of significant differences in the level of intracellular ATP between the tested strains during logarithmic growth, the activity of H^+^-ATPase was much lower in strain KS028 in comparison to the WT strain (Table 3, Figure 3). In the WT strain, *PMA1* expression was maintained at a higher level than in a mutant without ergosterol, especially in the early phase of growth (Figure 3E). Within 24 h of culture, however, the Pma1p level was lower in the KS028 mutant than in the WT strain (Figure 3D). Despite the lower level of Pma1p in the *ERG11* knockout mutant, and the delocalisation of the protein from the PM to the vacuole within 8 h of culture, we recorded a persistent presence of H^+^-ATPase for 24 h in the PM (Figure 3A). However, the H^+^-ATPase activity in late logarithmic phase is very low compared to the WT strain (Figure 3B). This indicates that the reason for reduced H^+^-ATPase activity is rather a lack of ergosterol in the PM, and subsequent reduced fluidity of the membrane, thus, the inability to maintain the correct location of the transporter, rather than the lack of energy for H^+^-ATPase function or a lower intracellular H^+^-ATPase level.

The upregulation of the *ERG11* gene is one of the established mechanisms of *C. albicans’* resistance to azoles [44]. Another type of azole resistance is their active efflux from cells in which ABC transporters are involved, mainly Cdr1p [45]. The decreased activity of Cdr1p in the ergosterol-free strain may result from a similar H^+^-ATPase lack of right localization of Cdr1p in the PM (Figure 4A), as well as from decreased PM potential caused by a decrease in H^+^-ATPase activity. 

We compare our results with the Erg11p external inhibition by treating *C. albicans* WT strain with fluconazole (Appendix A). An exposure of 24 h of the strain towards 4 μg/mL fluconazole resulted in an about four-fold increase in the *ERG11* gene expression (Appendix A). However, Erg11p activity was fully inhibited by fluconazole in this case, as indicated by the depletion of ergosterol and accumulation of lanosterol, 24-methyl-lanosterol and eburicol (Appendix A). When comparing the results with the sterol profile of *erg11Δ/Δ* strain (Table 2), it can be concluded that the external Erg11p inhibition differs from genetic inactivation. However, similarly to *erg11Δ/Δ* strain, fluconazole-treated WT strain displayed more rigid (Appendix A) and more depolarised (Appendix A) PM than in control conditions. Higher expression of *CDR1* gene was also present in case of fluconazole treatment (Appendix A), but to higher range than in case of *erg11Δ/Δ* strain (Figure 4C). We also noticed reduced efflux of R6G (Appendix A). The effect occurred most likely due to mislocalization of Cdr1p-GFP after fluconazole treatment (Appendix A). In *C. albicans* cells with reduced level of ergosterol, either under the action of azoles or genetic inactivation of *ERG11* gene, Cdr1p delocalises to vacuoles, so this may be a secondary, indirect effect of the antifungal action of these drugs. Constructed in our laboratory using YPD-nourseothricin selection (*SAT1* flipper targeted gene disruption) [22], the *C. albicans* mutant *erg11Δ/Δ* is viable under aerobic conditions, auxotrophic, and resistant to fluconazole and amphotericin B. In this mutant, varying concentrations and types of sterols and a lack of ergosterol contributed to a decrease in PM fluidity. We also observed in this a decreased level of ATP and different vacuole morphology than the WT strain, which is probably due to the dysfunction of the ergosterol-free membranes of the vacuole and mitochondria. The abnormal structure in *C. albicans erg11Δ/Δ* contributes to H^+^-ATPase and Cdr1 efflux pump delocalization to the vacuole by the early logarithmic growth phase and, as a result, to a decrease in membrane potential and increased sensitivity to xenobiotics. The conclusions resulting from our results are summarised in Figure 5. To our knowledge, this publication is the first to characterise the phenotype of *C. albicans erg11Δ/Δ* mutant obtained by the use of *SAT1* flipper system and to show the processes associated with azole resistance in cells lacking ergosterol.

## Figures and Tables

**Figure 1 microorganisms-07-00378-f001:**
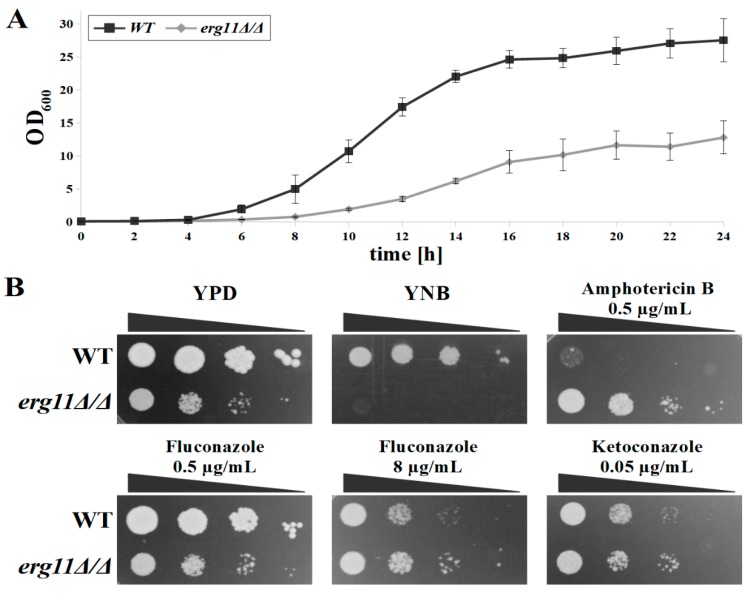
(**A**) Growth curves of *C. albicans* CAF2-1 wild type (WT) and KS028 (*erg11Δ/Δ*) strains in (2% glucose, 1% peptone, 1% yeast extract (YE)) medium (28 °C, 120 rpm); (**B**) Growth phenotypes of *C. albicans* CAF2-1 (WT) and KS028 (*erg11Δ/Δ*) strains after 48 h incubation at 28 °C. Growth phenotypes in the presence of amphotericin B, fluconazole and ketoconazole were performed in YPD medium.

**Figure 2 microorganisms-07-00378-f002:**
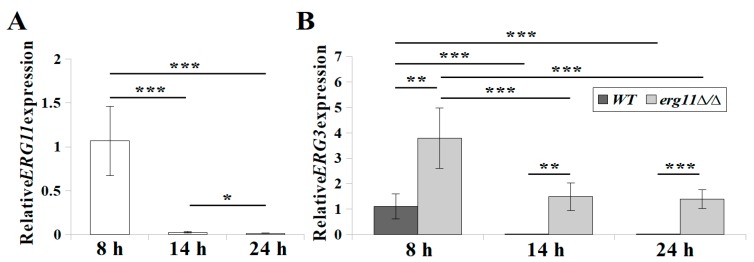
**(A**) Relative *ERG11* gene expression in *C. albicans* WT (CAF2-1) strain during growth (8, 14, and 24 h), not detected in the KS028 (*erg11Δ/Δ*) strain; (**B**) Relative *ERG3* gene expression in *C. albicans* WT and *erg11Δ/Δ* strains during growth (8, 14, and 24 h). Gene expression levels are reported as means of 2^−∆∆CT^ values (*n* = 6) ± SD; normalised to 1 after 8 h WT gene expression. Statistical analysis was performed by comparing expression during growth between *C. albicans* WT and *erg11Δ/Δ* strains (*, *p* < 0.05; **, *p* < 0.01; ***, *p* < 0.001).

**Figure 3 microorganisms-07-00378-f003:**
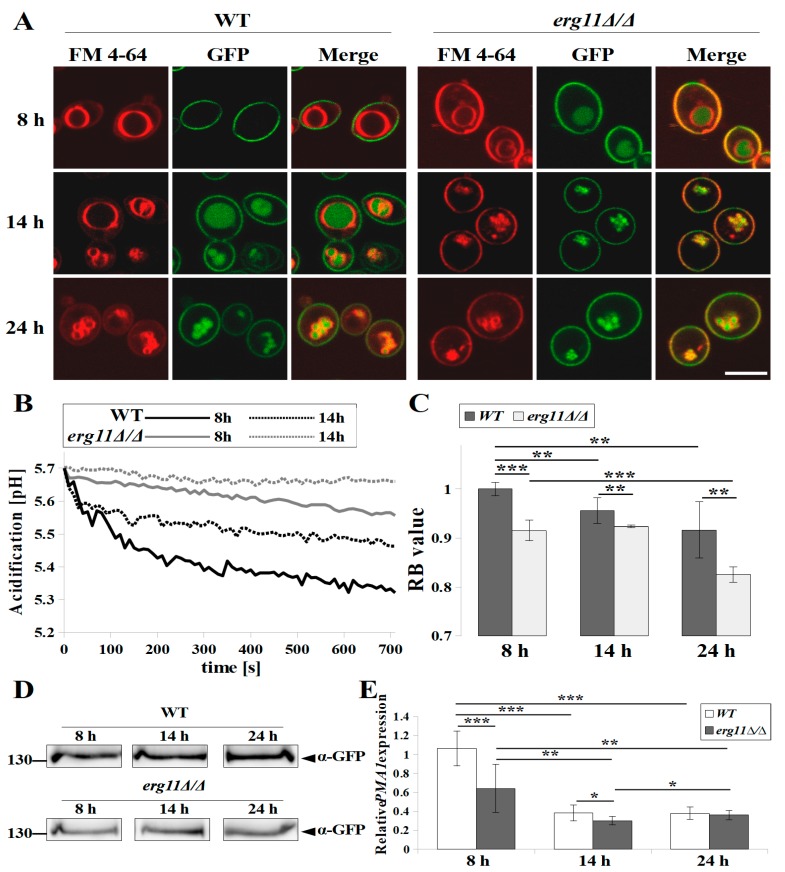
(**A**) Confocal micrographs of vacuolar membrane staining (FM 4-64) and subcellular localization of Pma1-GFP protein in *C. albicans* YHXW11 and KS045 (*PMA1-GFP, erg11Δ/Δ*) strains during growth (8, 14, and 24 h). A merged image of the GFP-tagged protein and FM 4-64 staining is shown in the third column. Scale bar = 5 µm. (**B**) Activity of Pma1p in *C. albicans* CAF2-1 (WT, black lines) and KS028 (*erg11Δ/Δ,* grey lines) strains after 8 h (solid lines) and 14 h (dashed lines) growth, expressed as acidification in time (0–720 s). Graphs are representative of three independent assays. (**C**) Plasma membrane potential (ΔΨ) expressed as RB values (means ± SD, *n* = 9) calculated from fluorescence spectra of di-4-ANEPPS incorporated into plasma membranes of *C. albicans* CAF2-1 (WT) and KS028 (*erg11Δ/Δ*) strains. (**D**) Immunoblot analysis of Pma1-GFP *p* levels in *C. albicans* YHXW11 (WT) and KS045 *(**PMA1-GFP*, *erg11Δ/Δ*) strains during growth (8, 14, and 24 h). The samples were resolved on 10% SDS-PAGE gel and probed by an anti-GFP antibody. Ponceau S staining was used as the loading control. Experiment is a representative of three independent assays and the presented conditions were resolved in the same gel and cut out into separate lines (Appendix A). (**E**) Relative *PMA1* gene expression in WT (CAF2-1) and KS028 (*erg11Δ/Δ*) strains during growth (8, 14, and 24 h). Gene expression levels as means of 2^−∆∆CT^ values (*n* = 6) ± SD; normalised to 1 for 8 h WT gene expression level. Statistical analysis in (**C**) and (**E**) compared data during different growth phases, and between *C. albicans* WT and KS028 (*erg11Δ/Δ*) strains (*, *p* < 0.05; **, *p* < 0.01; ***, *p* < 0.001).

**Figure 4 microorganisms-07-00378-f004:**
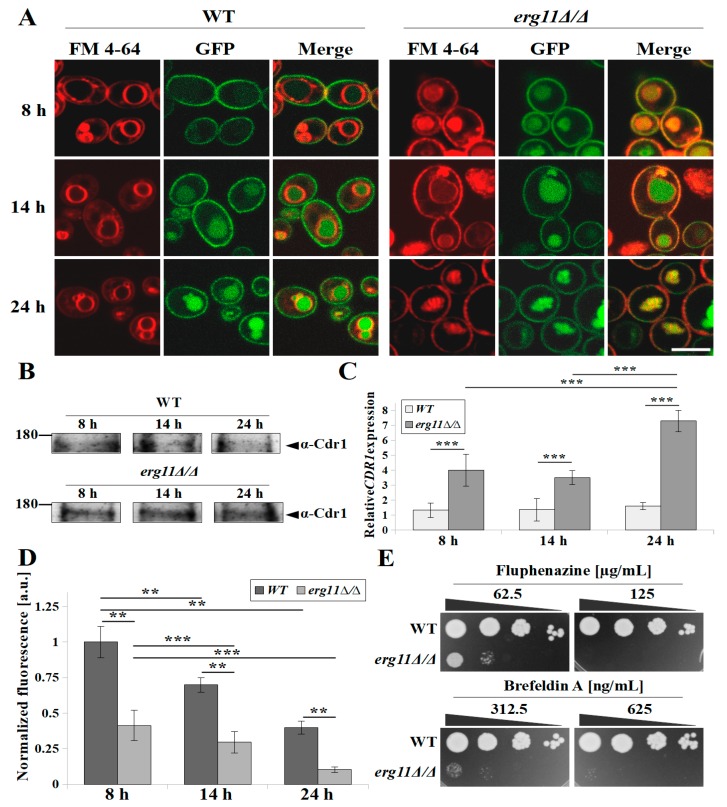
(**A**) Confocal micrographs of vacuolar membrane staining (FM 4-64) and subcellular localization of Cdr1-GFP protein in *C. albicans* strains ASCa1 and KS023 (*CDR1-GFP*, *erg11Δ/Δ*) during various growth phases (8, 14, and 24 h). A merged image of the GFP-tagged protein and FM 4-64 staining is shown in the third column. Scale bar = 5 µm. (**B**) Immunoblot analysis of Cdr1p levels in *C. albicans* CAF2-1 (WT) and KS028 (*erg11Δ/Δ*) strains during growth (8, 14, and 24 h). The samples were resolved in 6% SDS-PAGE gel and probed by an anti-Cdr1 antibody. Ponceau S staining was used as the loading control. Experiment is a representative of four independent assays and the presented conditions were resolved in the same gel and cut out into separate lines (Appendix A). (**C**) Relative *CDR1* gene expression in *C. albicans* CAF2-1 (WT) and KS028 (*erg11Δ/Δ*) strains during different growth phases (8, 14, and 24 h). Gene expression levels as means of 2^−∆∆CT^ values (*n* = 6) ± SD; normalised to 1 for 8 h WT strain gene expression level. (**D**) Cdr1p-dependent rhodamine 6G (R6G) efflux in *C. albicans* CAF2-1 (WT) and KS028 (*erg11Δ/Δ*) strains during growth (8, 14, and 24 h), shown as normalised (=1 for 8 h WT strain) fluorescence intensity of extracellular R6G (means ± SD, *n* = 3). (**E**) The growth phenotype of *C. albicans* CAF2-1 (WT) and KS028 (*erg11Δ/Δ*) strains after 48 h incubation at 28 ºC in the presence of fluphenazine (62.5 and 125 µg/mL) and brefeldin A (312.5 and 625 ng/mL). Statistical analysis in (**C**) and (**D**) compared data during different growth phases, or compared WT and *erg11Δ/Δ* strains (**, *p* < 0.01; ***, *p* < 0.001).

**Figure 5 microorganisms-07-00378-f005:**
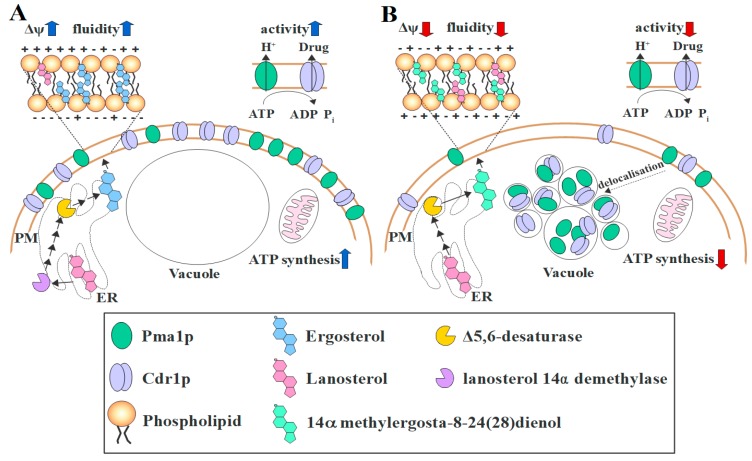
A general model of the influence of ergosterol on Cdr1p and Pma1p (H^+^-ATPase) localization; plasma membrane (PM) fluidity, and ATP level; red and blue arrows stand for lower and higher, respectively. **(A)** In a *C. albicans* wild type (WT) strain ergosterol is synthesised in the endoplasmic reticulum (ER) from lanosterol by several enzymes, including lanosterol 14α demethylase (Erg11p) and Δ5,6-desaturase (Erg3p). Ergosterol and lanosterol are incorporated into the PM and other membranes in the cell. Proper fluidity enables Pma1p and Cdr1p to localise into the PM, where the activity of both proteins depends on ATP, synthesised in the mitochondria. H^+^-ATPase modulates the potential of the PM (Δψ). (**B**) Deletion of the *ERG11* gene results in non-specific modifications of lanosterol. The final step is catalysed by Erg3p and leads to the incorporation of 14α methylergosta-8-24(28)dienol with lanosterol in the PM. Abnormal PM composition and rigidity disables Pma1p and Cdr1p from proper localization and these proteins delocalise into vacuoles, which are not fused in *erg11Δ/Δ*. Delocalization and reduced ATP level lead to reduced activity of Cdr1p and Pma1p. The potential of PM is reduced due to low Pma1p activity.

**Table 1 microorganisms-07-00378-t001:** *C. albicans* strains used in this study.

Strain	Parent	Genotype	Reference
CAF2-1		*ura3Δ::imm434/URA3*	[9]
ASCa1		*ura3Δ::imm434/ura3Δ::imm434* *CDR1/CDR1-yEGFP-URA3*	[10]
YHXW11		*ura3Δ::imm434/ura3Δ::imm434* *PMA1/PMA1-GFPγ-URA3*	[11]
KS018	ASCa1	*ura3Δ::imm434/ura3Δ::imm434* *CDR1/CDR1-yEGFP-URA3* *ERG11/erg11Δ::SAT1-FLIP*	This study
KS021	KS018	*ura3Δ::imm434/ura3Δ::imm434* *CDR1/CDR1-yEGFP-URA3* *ERG11/erg11Δ::FRT*	This study
KS023	KS021	*ura3Δ::imm434/ura3Δ::imm434* *CDR1/CDR1-yEGFP-URA3* *erg11Δ::SAT1-FLIP/erg11Δ::FRT*	This study
KS014	CAF2-1	*ura3Δ::imm434/URA3* *ERG11/erg11Δ::SAT1-FLIP*	This study
KS020	KS014	*ura3* *Δ* *::imm434/URA3* *ERG11/erg11* *Δ* *::FRT*	This study
KS028	KS020	*ura3Δ::imm434/URA3* *erg11Δ::SAT1-FLIP/erg11Δ::FRT*	This study
KS043	YHXW11	*ura3Δ::imm434/ura3Δ::imm434* *PMA1/PMA1-GFPγ-URA3* *ERG11/erg11Δ::SAT1-FLIP*	This study
KS044	KS043	*ura3Δ::imm434/ura3Δ::imm434* *PMA1/PMA1-GFPγ-URA3* *ERG11/erg11Δ::FRT*	This study
KS045	KS044	*ura3Δ::imm434/ura3Δ::imm434* *PMA1/PMA1-GFPγ-URA3* *erg11Δ::SAT1-FLIP/erg11Δ::FRT*	This study

**Table 2 microorganisms-07-00378-t002:** Sterols (µg/mg dry mass of isolated lipids, means ± SD, *n* = 3) in *C. albicans* CAF2-1 (WT) and KS028 (*erg11Δ/Δ*) strains. Statistical analysis was performed in accordance to µg/mg values after 8 h of culture (**, *p* < 0.01; ***, *p* < 0.001); General polarization values (GP; means ± SD, *n* = 6) of Laurdan incorporated into plasma membranes of *C. albicans* CAF2-1 (WT) and KS028 (*erg11Δ/Δ*) strains. Statistical analysis was performed in accordance to GP values after 8 h for the WT strain (**, *p* < 0.01; ***, *p* < 0.001).

Strain	Time of Culture (h)	Ergosterol	Lanosterol	14α Methylergosta-8-24(28)dienol	GP
WT	8	39.11 ± 2.3	8.48 ± 0.2	-	-0.41 ± 0.03
	14	65.17 ± 1.1 ***	5.4 ± 0.6 **	-	-0.32 ± 0.06 ***
	24	6.89 ± 0.2 **	3 ± 0.2 ***	-	-0.07 ± 0.05 ***
*erg11Δ/Δ*	8	-	22.5 ± 1.3	23.7 ± 2.4	-0.18 ± 0.08 ***
	14	-	21.6 ± 2.4	46 ± 3.7 **	-0.17 ± 0.03 **
	24	-	15.6 ± 1.6*	61.8 ± 4.3**	-0.07 ± 0.008***

**Table 3 microorganisms-07-00378-t003:** Intracellular level of ATP [nM] per 1.4 × 10^5^ cells (means ± SD, *n* = 6), measured by luciferase luminescence in *C. albicans* WT (CAF2-1) and KS028 (*erg11Δ/Δ*) strains during growth (8, 14, and 24 h). Statistical analysis was performed in accordance to GP value for 8 h WT strain (*, *p* < 0.05; **, *p* < 0.01).

Strain	Time of Culture (h)	ATP (nM/1.4 × 10^5^ cells)
WT	8	29.3 ± 5.4
	14	26.6 ± 2
	24	13.5 ± 1.5 **
*erg11Δ/Δ*	8	27.7 ± 2
	14	22 ± 1.5 *
	24	2.8 ± 0.3 **

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
