# Peer review of "A Crucial Role for Ergosterol in Plasma Membrane Composition, Localisation, and Activity of Cdr1p and H+-ATPase in Candida albicans"

_microorganisms, 2019, doi:10.3390/microorganisms7100378_

Round 1

Reviewer 1 Report

The authors of microorganisms-586208 performed targeted gene disruption of ERG11 in Candida albicans and rigorously assessed its impact on several cellular processes. However, the aim of the manuscript is not clear. Translation of the results of experiments with erg11Δ/Δ cells into conclusions regarding enzyme inhibition seems too far-reaching. Please see the detailed comments below.

MAJOR COMMENTS

1. The aim of the study has to be clearly stated. Currently, there is no information on what questions the authors were trying to answer.

2. Page 2, lines 7-9 have to be rewritten or deleted:

ERG11 appears to be an excellent target for antifungal therapy” – One may argue that it is not so perfect target since human cells also contain sterol 14-alpha demethylase… and here comes the azole toxicity…

“Blocking its function may significantly interrupt existing mechanisms of resistance in C. albicans” – I do not understand what message the authors want to deliver. Azoles, which target Erg11, are known for a long time. Thus, how come blocking ERG11 can help overcome azole resistance?

3. AFST of the strains should be performed with a standardized broth microdilution method (EUCAST or CLSI). Please determine azole MIC values for each strain in order to compare their susceptibility. Moreover, it would enable detection of trailing growth effect that the authors probably observed (Page 6, lines 27-29).

4. In order to draw conclusions about the enzyme inhibition by drugs, a series of experiments (same as performed for WT and erg11Δ/Δ) should be performed for WT cells exposed to azoles. Currently, the authors do not provide any proof that the observed changes are the same in erg11Δ/Δ and azole-treated cells. A sentence like “If in C. albicans cells with reduced level of ergosterol, under the action of azoles, Cdr1p delocalises to vacuoles, this may be a secondary, indirect effect of the antifungal action of these drugs” (Page 14, lines9-11) is too hypothetical.

Moreover, relevant literature should be discussed in more detail, e.g.:Sanglard D, Ischer F, Parkinson T, Falconer D, Bille J. 2003. Antimicrob Agents Chemother. 47(8):2404-12. “Upregulation of both ERG3 and ERG11 in mutants lacking either gene had an effect comparable to that from the inhibition of sterol biosynthesis by the addition of azoles or other ergosterol biosynthesis inhibitors. This phenomenon has been described (…) in C. albicans exposed to itraconazole” => 1) De Backer, M. D., T. Ilyina, X. J. Ma, S. Vandoninck, W. H. Luyten, and H. Vanden Bossche. 2001. Antimicrob. Agents Chemother. 45:1660-1670. 2) Henry, K. W., J. T. Nickels, and T. D. Edlind. 2000. Antimicrob. Agents Chemother. 44:2693-2700

MINOR COMMENTS

Page 1, lines 28-31: what about candidemia= invasive candidiasis?

Page 1, line 42: should be “overexpression” not “overproduction”

Page 2, line 35: should be “mL” not “mLs”

Page 4, line 2: should be “heterozygous” not “heterozygotous”

Page 4, line 10: 10-3 cells/mL doesn’t sound right. What concentrations were prepared?

Page 4, line 35: run length and temperature missing

Page 5, line 15: should be “CAF2-1” not “CAF201”

Page 13, line 25: remove “knock-out”

Author Response

Dear Reviewer,

we send our answers and explanations to ms. We also send corrected ms and supplemented supplementary materials.

Sincerelly

Anna Krasowska

Reviewer 2 Report

The methods used for S.Cerevisiae and C. glabrata as comparison for C. Albicans should be detailed in the experimental methods also,not only in the results part, to simplify the understanding of the readers.

The quality of article is good, need just minor revision

1. Please reformulate the paragraph from page 2, lines 7-9. it is not clear: “ERG11 appears to be an excellent target for antifungal therapy”

2. Please replace mLs with mL at page 2, line 3

3. Please replace “heterozygotous” with “heterozygous” at page 4, line 2.

4. Please give us details about the results from the experiments for the WT cells exposed to azoles.

5. Double check References.

Author Response

(The authors gave the same response as above.)

Round 2

Reviewer 1 Report

The authors provided satisfactory responses to most of my comments. However, several aspects still need to be corrected.

Neither Figure S4 nor S5 is present in the Supplementary Material. Thus, it is not possible to evaluate the claims regarding the effects of fluconazole on the WT cells.

Page 2, lines 9-10: still not clear. From the authors' response to my previous comments, I deduct that what they mean is blocking ERG11 expression, not function (azoles block the enzyme function). A concept of a "genetic drug" (mentioned in the response to comments) should be better explained.
